# Muscle Activity Detectors—Surface Electromyography in the Evaluation of Abductor Hallucis Muscle

**DOI:** 10.3390/s20082162

**Published:** 2020-04-11

**Authors:** Kamila Mortka, Agnieszka Wiertel-Krawczuk, Przemysław Lisiński

**Affiliations:** 1Department of Rehabilitation and Physiotherapy, Poznan University of Medical Sciences, ul. 28 Czerwca 1956 nr 135/147, 60-545 Poznań, Poland; plisinski@vp.pl; 2Department of Pathophysiology of Locomotor Organs, Poznan University of Medical Sciences, ul. 28 Czerwca 1956 nr 135/147, 60-545 Poznań, Poland; wiertel-krawczuk@wp.pl

**Keywords:** surface electromyography, electroneurography, abductor hallucis muscle, toe-spread-out exercise, hallux valgus

## Abstract

Despite the high availability of surface electromyography (sEMG), it is not widely used for testing the effectiveness of exercises that activate intrinsic muscles of foot in people with hallux valgus. The aim of this study was to assess the effect of the toe-spread-out (TSO) exercise on the outcomes of sEMG recorded from the abductor hallucis muscle (AbdH). An additional objective was the assessment of nerve conduction in electroneurography. The study involved 21 patients with a diagnosed hallux valgus (research group A) and 20 people without the deformation (research group B) who performed a TSO exercise and were examined twice: before and after therapy. The statistical analysis showed significant differences in the third, most important phase of TSO. After the exercises, the frequency of motor units recruitment increased in both groups. There were no significant differences in electroneurography outcomes between the two examinations in both research groups. The TSO exercise helps in the better activation of the AbdH muscle and contributes to the recruitment of a larger number of motor units of this muscle. The TSO exercises did not cause changes in nerve conduction. The sEMG and ENG are good methods for assessing this exercise but a comprehensive assessment should include other tests as well.

## 1. Introduction

Surface electromyography (sEMG) is, next to dynamometry, ultrasound and magnetic resonance imaging, a commonly used research tool for assessing muscles. The benefits of this type of muscle evaluation using surface sensors have been implemented in many research fields such as medicine, rehabilitation, sport, human-computer interactions, prosthetics and ergonomics. Surface electromyography is well-suited for an assessment of the effects of physiotherapeutic [1], surgical and even pharmacological treatment [2]. It is also widely utilized in dynamic gait analysis [3,4], in competitive sports for testing muscle fatigue after prolonged effort, and in ergonomics tests for assessing overloaded muscles during work.

Additionally, modern technology and the minimization of surface electromyography devices have allowed the use of this method in biofeedback training. It is a method of training muscles by creating new feedback systems as a result of the conversion of myoelectrical signals in muscles into visual and auditory signals [5]. The sEMG biofeedback has been used for general relaxation training, stroke rehabilitation and the treatment of pain [6]. These kinds of devices can help to “down-train” elevated muscle activity or to “up-train” weak, inhibited, or paretic muscles [5].

The indisputable advantage of sEMG is its complete non-invasiveness. It is a painless and easy to use test for children. Another advantage is the ability to simultaneously register potentials from many muscles, as well as the ease of application in dynamic tests. In turn, the main limitation of this method is signal sensitivity to internal and external factors [7]. Due to the overwhelming benefits of this method, recent decades have contributed to the creation and dissemination of low-cost sEMG devices, some of which are positively validated in scientific research [8,9,10].

Despite the high availability of surface electromyography, it is not widely used to test the effectiveness of exercises that activate intrinsic muscles of the foot in people with hallux valgus deformation. Most of them refer to the evaluation of such exercises using EMG in healthy people. Such an assessment was made, for example, by Goo et al. [11], Jung et al. [12] and Heo et al. [13]. These researchers studied changes in the activity of the abductor hallucis muscle (AbdH) among asymptomatic people who performed various intrinsic muscle exercises. The number of groups of two of these studies was relatively small: 11 subjects in the Goo et al. [11] study and 12 participants in the study by Heo et al. [13]. None of these all three studies had a control group in their design. These researchers compared the effects of different exercises with each other [11,12,13] or the effects of the same exercise performed in different positions [11,12]. All these studies assessed AbdH muscle activity during the particular exercise, possibly in various positions, but did not examine the effectiveness of these exercises in relation to the long duration of these exercises, i.e., repeatedly performing them.

In the literature only a few researchers [14,15] have assessed selected exercises in patients with hallux valgus and their effect on the activity of the AbdH muscle although many studies [16,17,18,19] indicate that this muscular structure is significantly involved in hallux valgus deformation. These studies [16,17,18,19] report muscle strength imbalance between the abductor and adductor muscles, but it has not been proved, whether it is a reason or the result of joint deformity. Additionally, some researchers [17] suggested that the bioelectrical activity of the AbdH muscle during abduction in the first metatarsophalangeal joint is significantly decreased in patients with hallux valgus compared with a control group.

There is only one study [14] using the EMG method and regarding the evaluation of the effects of intrinsic muscle exercises in patients with hallux valgus. This research by Kim et al. [14] refers to the activity of AbdH muscle. The purpose of the research mentioned above [14] was to compare the effects of the toe-spread-out exercise (TSO) with another exercise—short foot exercise (SF). The authors of this study did not analyze EMG results before and after exercise, but only compared them between exercises. Therefore, based on their research, it cannot be clearly stated whether these exercises are effective in activating the AbdH muscle in patients with hallux valgus.

For that reason, we decided to investigate this area and to evaluate in patients with hallux valgus deformity one of AbdH muscle exercise—the toe spread out exercise (TSO) using surface electromyography. Due to the fact that no one has ever used electroneurography (ENG) to assess nerve conduction as a result of hallux valgus exercises, we decided to use this research tool additionally.

This study aimed to assess the effect of the TSO exercises on the amplitude and frequency of motor unit activation during maximal contraction of AbdH analyzed in surface electromyography in patients with hallux valgus and without the deformity. An additional objective was the assessment of tibial nerve motor fibers function in electroneurography. 

The novelty of this study was the assessment of the effectiveness of TSO exercises on AbdH muscle function, where the commonly used method for assessing bioelectric muscle activity (sEMG) has not yet been widely discussed in the case of hallux valgus.

## 2. Materials and Methods

### 2.1. Design of the Study

The design of this research used an interventional model. Two research groups were created (research group A—with a diagnosis of hallux valgus and research group B—without the deformity), which received therapeutic intervention of the TSO exercise performed unilaterally. Additionally, a small control group with hallux valgus deformity, which did not undergo any therapy, was created. This aimed to check if any changes of the analyzed parameters are in fact the result of the exercises. All participants were examined in the same way twice: before and after the therapy (research groups) or at an interval of 14 days (control group). The assessment of the effects of therapeutic intervention (TSO exercise) was based on neurophysiological studies: surface electromyography and electroneurography. The design of this study is presented in the Figure 1.

### 2.2. Materials

Finally, research group A included 21 feet of 21 patients aged 22–78 (mean 44.43 ± 17.87) with a diagnosis of hallux valgus (HVA 16°–43°, FIA 6–19°). The characteristics of the valgus of the exercised foot in research group A are shown in Table 1. Research group B encompassed 20 feet without deformity of 20 patients aged 21–69 (mean 35.50 ± 12.84). The control group consisted of 10 feet with hallux valgus of 5 patients aged 32–69 (mean 52.8 ± 13.86) with the hallux valgus deformity (HVA 16–32°, FIA 8–13°).

### 2.3. Inclusion and Exclusion Criteria

The Manchester scale and the clinical examination performed with the use of a goniometer were the basis for the inclusion criteria to the research groups and to the control group. It was decided to use such screening methods due to their non-invasiveness and simplicity of implementation. The Manchester scale is based on 4 standard photos (A, B, C and D), which allow assessing the severity of the hallux valgus. The investigator evaluates which picture resembles the subject most closely. Picture “A” presents a lack of distortion whereas the other (“B”, “C” and “D”) shows the increasing degrees of the valgus. Score “B”, “C” or “D” in the Manchester scale, as well as the hallux valgus angle above 15° (measured with a goniometer and then confirmed in the X-ray examination) was the basis for the inclusion of the patient to research group A and control group. The inclusion criteria of the patients to research group B was the “A” score in the Manchester scale, as well as the hallux valgus angle equal or lower than 15° (measured with a goniometer).

In order to standardize the group and to exclude patients with other illnesses, which could have some impact on the weight-bearing patterns of the foot, the following exclusion criteria to research group A, research group B and the control group were established: a serious injury or a history of lower limb surgery, presence of low back pain on the background of discopathy with ventral spinal roots conflict (based on the ENG study), neuropathy, myopathy, rheumatoid arthritis, psoriatic arthritis, connective tissue diseases, a history of strokes or other neurological diseases.

After the patient was included in research group A or in the control group, the hallux deformities were assessed based on X-ray images taken in the anterior–posterior projection in weight-bearing conditions in a standing position. The hallux valgus angles (HVA) and the first intermetatarsal angles (FIA) on all images were designated and measured by one experienced radiologist. The result above 15 degrees of the HVA was confirmation of the hallux valgus diagnosis and the result equal or below this value was an exclusion criterion from research group A or from control group.

### 2.4. Methods

The effects of the TSO exercises were evaluated using a comparison of the results of two examinations: before and after the therapeutic intervention (in the research group A and B) or simply at an interval of 14 days (in the control group).

The therapeutic intervention was based on performing TSO exercise for a period of 14 days under the supervision of a qualified physiotherapist. This TSO exercise starts in the sitting position with the knee joint and hip bent at 90 degrees and it consists of 3 consecutive phases. The first phase is dorsiflexion of the toes with keeping the metatarsal heads and the heel on the ground. The second is adding to this position the movement of the fifth toe down and in a lateral direction. In turn, the third phase is based on moving the big toe down and abduction. The exerciser should maintain the final position for 5 s. The participants repeated the whole sequence 200 times a day.

To examine the effects of TSO exercises, a surface electromyography and electroneurography were used. The methodology abandoned the use of elemental electromyography for several reasons. Elemental electromyography is an invasive test assessing muscle function with a needle electrode. It is mainly used in the diagnosis of neuromuscular diseases and analyzes in detail the parameters of motor unit action potentials (MUAP). The application of a needle electrode and the accompanying pain could affect the bioelectric activity of the muscle, such as limiting the range of motion and disturbed the frequency of motor unit recruitment during the TSO exercise sequences. Hence, non-invasive sEMG was chosen as the diagnostic method in this study. Additionally, this method enabled the assessment of bioelectric activity on-line, which allows patients to understand the principles of TSO exercises as well as to control their correct performance (biofeedback convention).

The sEMG and the ENG examinations were performed with the use of the KeyPoint System (Medtronic A/S, Skovlunde, Denmark). The appropriate set of electrodes (surface electrodes, grounding electrode and bipolar stimulating electrode) depending on the type of neurophysiological studies (sEMG or ENG) was applied. During ENG examination, the patient was lying on a couch, in a comfortable position on his/her back, in full muscle relaxation. The sEMG studies were performed in a sitting position with the feet on the floor.

In surface electromyography, it was decided to assess the activity of the AbdH muscle, because it is significantly involved in the hallux valgus deformation. For the examination of the AbdH activity the standard disposable Ag/AgCl surface electrodes with an active surface of 5 mm^2^ were used. The active electrode (cathode) was placed on the belly of AbdH muscle and the reference electrode (anode) was located 3 cm distally from the active electrode. In this sEMG examination, the time base on 80 ms/D and sensitivity of the recording of 0.5 mV/D were set. Moreover, 10 kHz upper and 20 Hz lower filters of the recorder amplifier were used. For reduced electrical resistance between the electrodes and the skin, the gel was applied to each electrode. Therefore, impedance did not exceed 5 kΩ.

The recordings from the AbdH muscle were performed during three phases of the TSO exercise. The analysis was based on the following parameters: amplitude of motor unit action potential (MUAP) measured in mV, and the frequency pattern of the MUAP recruitment of AbdH muscle activity. In order to obtain more objective results, the sEMG examinations with maintaining each of the three phases of the TSO exercise were performed three times. The first recording was treated as a training trial. The other two trials were taken into account to evaluate the amplitude and the frequency pattern. Only the recording with the highest amplitude and the most interference pattern was treated as the result of the most effective recruitment of the MUAPs. After the marking of the MUAP amplitude peaks, the KeyPoint System measured the minimum and maximum amplitude values automatically. An ‘on-line’ assessment of the frequency of the MUAP recruitment during a maximum AbdH muscle contraction was a subjective visual evaluation made by one experienced neurophysiologist. This assessment was based on the classification presented by Buchthal et al. [20] and by Stalberg and Falck [21] and included the following frequency patterns: interference, intermediate, poor and straight. The details of the sEMG examination are presented in Figure 2.

In order to examine the effects of TSO exercise on nerve conduction, the electroneurography was used to assess the motor fibers function of the tibial nerve, which innervate the AbdH muscle. The ENG examination was conducted according to the methodology of Stalberg and Falck [21]. For the recording of the compound muscle action potentials (CMAP) from the tibial nerve, the surface electrodes were applied in the same location as in the sEMG examination. The ground electrode was located on the lower leg. In order to stimulate the tibial nerve, a bipolar stimulation electrode was used, which was moisturized with a saline solution (0.9% NaCl). The position on the skin where the ground electrode, active electrode and reference electrode were placed was disinfected with a 70% alcohol solution, which reduced the skin’s resistance. A gel was applied to each electrode (except the stimulation electrode), which reduced the electrical resistance between the electrode and the skin. Impedance did not exceed 5 kΩ.

The tibial nerve was stimulated in two points: below the medial malleolus and in the popliteal fossa. The bipolar stimulation electrode delivered single rectangular stimuli with a duration of 0.2 ms at a 1 Hz frequency. The intensity of the stimuli ranged from 30 mA to the value evoking the supramaximal CMAP (with the highest value of amplitude and the shortest latency value). In the ENG examination the time base was set on 5 ms/D and sensitivity of the recordings on 2 mV/D. Moreover, 10 Hz upper and 10 kHz lower filters of the recorder amplifier were used.

The analysis of the CMAP from the tibial nerve was based on the following parameters: amplitude of the negative potential (the beginning of the negative deflection—peak of the potential—isoelectric line—measured in mV ), latency measured from the beginning of the application of the stimulus (with a visible artifact) to the beginning of the negative deflection from the isoelectric line (measured in ms) and segmental conduction velocity at the level of the lower leg (measured in m/s). The details of the ENG examination and sample results of the evaluated parameters are presented in Figure 3.

Ethical considerations of the study were compliant with the Helsinki Declaration. Approval was also granted by the Bioethical Committee of the University of Medical Sciences, which is confirmed by the resolution numbered 64/16.

### 2.5. Statistical Analysis

All data analyses were conducted with the use of the CytelStudio, version 10.0, created in January 16 2013 (CytelStudio Software Corporation, Cambridge, MA, USA), and StatSoft, Inc. (2014). STATISTICA (data analysis software system), version 12 from www.statsoft.com.

Normality was determined by the Shapiro–Wilk test. The differences between the groups were estimated using the Wilcoxon test for non-normally distributed data or a paired Student’s *t*-test for normally distributed data.

A *p*-value below 0.05 was judged to be statistically significant.

## 3. Results

The AbdH muscle activity during the three phases of the TSO exercise was analyzed in each group, taking into account the amplitude and the frequency pattern. In research group A, the Wilcoxon test showed that there was a significant difference (*p* = 0.01) in the frequency pattern in the exercised foot between examinations before and after therapy, but only in the third phase of the TSO exercise. After the exercises, the frequency was higher and more similar to an interference pattern. Figure 4 shows examples of sEMG results for a selected patient from research group A.

Whereas in research group B, the statistical analysis using the Wilcoxon test confirmed the significant difference in the exercised foot in the second phase considering the amplitude (*p* = 0.04) and frequency pattern (*p* = 0.01). After the exercises, the frequency increased and was more similar to interference. The amplitude changes in the second phase of TSO are presented in Figure 5. In the third phase of TSO exercise the Wilcoxon test showed that there was a significant difference only in the frequency pattern (*p* = 0.02), but not in the amplitude (*p* = 0.2). After the exercises, the frequency increased and was more similar to interference.

A further statistical analysis included a comparison of the ENG parameters (CMAP from the tibial nerve) before and after therapy in the exercised foot (research group A and B). In research group A and B there were no significant differences in amplitude (measured below the medial malleolus and in the popliteal fossa), latency (measured below the medial malleolus and in the popliteal fossa) and conduction velocity between two examinations, which is presented in Table 2. 

The neurophysiological parameters of the two examinations (at an interval of 14 days) were compared in the control group as well. No statistically significant differences were found between the parameters of the ENG and sEMG in this group.

## 4. Discussion

In our study we primarily focused on an assessment of the activity of the AbdH muscle, as a muscle structure that is significantly involved in hallux valgus deformation, which was reported in previous years [16,17,18,19]. The design of this study is similar to the study of Kim et al. [14]. Both studies used a surface electromyography to assess the effectiveness of the TSO exercise in AbdH muscle activation in patients with hallux valgus. In our study we evaluated the bioelectrical activity of this muscle in patients with hallux valgus twice: before the TSO exercises and after 2 weeks of performing them. In addition, we compared muscle activity in people without the distortion as well. In the study of Kim el al. [14] the activation of the AbdH muscle was assessed once, after 2 weeks of TSO. The purpose of their research was to compare the effects of the TSO exercise with another exercise—short foot exercise (SF)-without comparing the effects before and after therapy or results with a group of people without hallux deformities. In addition, these authors compared the abovementioned exercise models in terms of AbdH muscle activity in relation to the adductor muscle (AddH) activity and measured the angle of the MTP joint.

Other papers that also evaluate the TSO exercise differ significantly from our study in terms of design and methodology. For example, Gooding et al. [22] used magnetic resonance to evaluate the effects of TSO. In addition, their study was based on healthy people without foot deformities, and its purpose was to analyze changes in the activities of intrinsic muscles of the foot after a series of 40 repetitions of four different exercises. Therefore, the kinesiotherapy intervention was the same as in our study, but its duration and intensity vary significantly.

Considering the differences described above in the projects, objectives and methodology, it is difficult to directly compare the results of our research with the results of other authors. Our study and the publications mentioned earlier assessed the same TSO exercise, but they differ in the duration of the exercise and characteristics of the control groups. However, in the further part of the discussion, we will try to compare the results obtained by us with the results of others.

The aim of this study was to assess AbdH muscle activity by using the electromyography and electroneurography. Nevertheless, muscle activity should be considered both in terms of muscle structure and strength. Comprehensive muscle assessment should include other tests and diagnostic methods, for example based on dynamometry, magnetic resonance imaging or ultrasonography. Unfortunately, there are no studies that allow reference to such a comprehensive analysis of the AbdH muscle in patients with hallux valgus.

However, our study assumed that changes may occur in the sEMG examination and the results confirmed this supposition. The third phase of the TSO exercise seems to be the most important, because it is the most activating phase of the AbdH muscle and the previous phases are just consecutive steps in reaching it. Therefore, a further analysis of the results will focus on the results of this phase. The frequency pattern of MUAPs in this phase was significantly higher after exercise in the patients with hallux valgus (research group A), which confirmed the Wilcoxon test (*p* = 0.01). Similar changes were observed in the patients without any deformity (research group B). The Wilcoxon test showed significant differences in the results of examination before and after therapy. After the exercises, the frequency increased and was more similar to interference. These changes could not have been caused by the passage of time or a measurement error because in the control group there were no differences in the analyzed parameters.

Our hypothesis assumed also changes after the TSO exercises in MUAPs amplitude. However, the statistical analysis using Wilcoxon test confirmed the significant differences (*p* = 0.04) in this parameter only in the second phase of TSO considering examinations before and after exercises. In the third, most important phase of this exercise, there were no significant differences (*p* = 0.2) in the amplitude. The amplitude is related to the net of the motor unit activity: the recruitment and the discharge rates of the active motor units, which is emphasized by Farina et al. [23]. However, these authors indicate that EMG amplitude is influenced by such factors as electrode location, thickness of the subcutaneous tissues, distribution of motor unit conduction velocities and the detection system used to obtain the recording. Nevertheless, the results of the changes in frequency in our study suggest that after exercises the number of recruited motor units increased. Farina et al. [23] suggest according to Bernardi et al. [24,25] that shifts in the mean or median frequencies should reflect the recruitment of progressively larger and faster motor units.

These results do not compare with the results of Kim et al. [14], because they assessed only once the EMG amplitude of the AbdH muscle during each exercise, expressed as a percentage of the maximum voluntary isometric contraction (%MVIC). They did not take into account the frequency pattern.

The results of our study regarding the increase in AbdH muscle activity in the form of a change in frequency pattern after exercise are consistent with the results of Gooding et al. [22], who also noted an increase in its magnetic resonance image. In addition, in the study of Gooding et al. [22] the TSO exercise showed the largest increase in activation expressed as a percentage in relation to the abductor muscle of the fifth toe (35.2%), the oblique head of the adductor hallucis (31.5%) and the flexor of the fifth toe (30.2%). In turn, this increase for the AbdH muscle was 18.9%.

The assumptions of our study included an assessment of the activity of one muscle, the AbdH muscle, without comparing it with the activity of the AddH muscle, which Kim et al. [14] and Gooding et al. [22] did. Comparing these two researches, it should be stated that their results are slightly different. In the study of Kim et al. [14], the exercise of TSO showed greater activation of the AbdH muscle compared to AddH. The results of Gooding et al. [22] also refer to these muscles, but the purpose of the study was not to compare their activity during the exercise, but to evaluate the increase in activity before and after therapeutic intervention. In the initial study, the activity of these two muscles in the examined group of people without deformation was similar. The differences between the results of these two researches may result from other examined groups of patients: people with hallux valgus (Kim et al) and without this deformation (Gooding et al).

In turn, the results of the ENG of our study indicate that the exercises performed for 14 days did not cause any changes in the tibial nerve function. No significant differences in the amplitude and latency of the CMAP and conduction velocity in the tibial nerve were observed in exercising patients with hallux valgus, considering both the pre- and post-treatment examination. The ENG results are similar in research group B with people exercising without hallux deformation. So far, there are no references in the medical literature with which one could compare the above results of the impact of the TSO exercise on the parameters of electroneurographic examination. It can be concluded that the hallux valgus is associated only with the muscular imbalance between muscles of different functions (AbdH muscle vs. AddH muscle) with preserved normal nerve conduction in the tibial nerve.

## 5. Conclusions

Fourteen days of TSO exercises can result in changes of the frequency pattern of AbdH muscle activity in patients with hallux valgus and without deformity. The TSO exercise helps in the better activation of the AbdH muscle and contributes to the recruitment of a larger number of motor units of this muscle. The TSO exercises performed for 14 days did not cause changes in nerve conduction. Electromyography and electroneurography are good methods for assessing the toe spread out exercise but a comprehensive assessment should include other tests as well. Further studies are needed for an assessment of the TSO exercise after a longer period of therapy and with a larger number of patients.

## Figures and Tables

**Figure 1 sensors-20-02162-f001:**
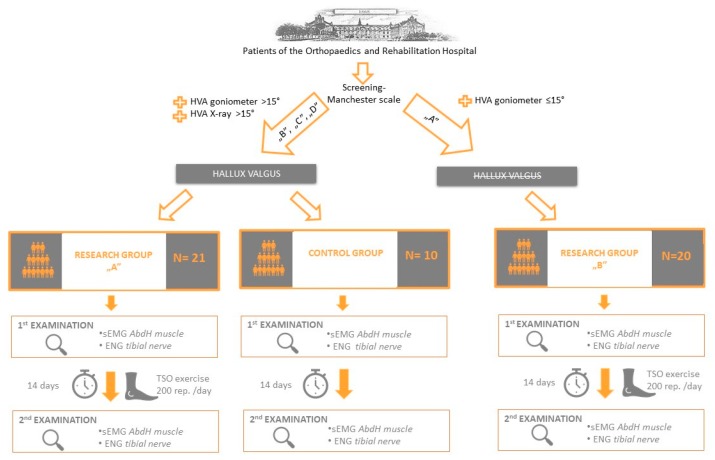
The design of the study (HVA—hallux valgus angle, 1st examination—first examination before exercise, 2nd examination—second examination after exercise, sEMG—surface electromyography, ENG—electroneurography, AbdH muscle—abductor hallucis muscle, TSO—toe spread out exercise, rep.—repetitions).

**Figure 2 sensors-20-02162-f002:**
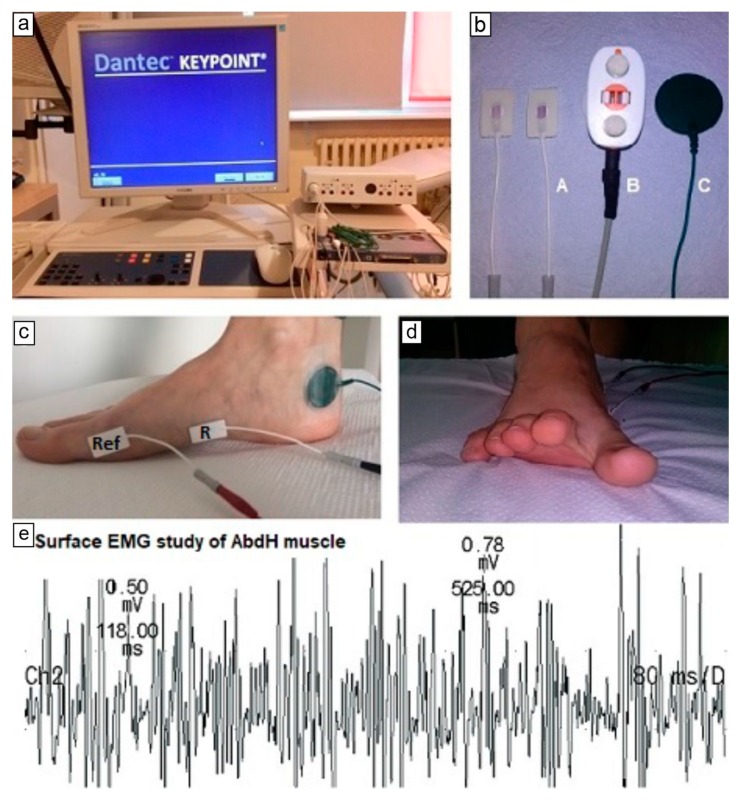
Surface EMG study of abductor hallucis muscle. (**a,b**)—KeyPoint System and set of electrodes; (**c**) application of electrodes on the AbdH muscle (R—recording electrode, Ref—reference electrode) and (**d**) the third phase of TSO exercises and (**e**) recording of AbdH muscle activity during maximal contraction.

**Figure 3 sensors-20-02162-f003:**
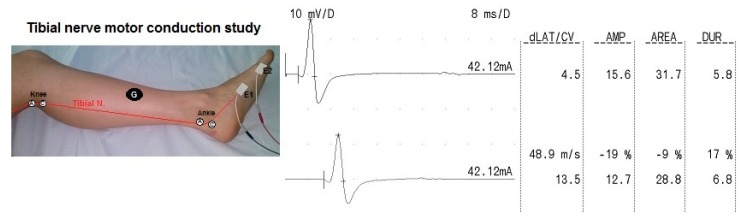
Placement of the electrodes in the electroneurographic examination of the tibial nerve and sample results of the evaluated parameters (E1—active recording electrode; E2—reference electrode; stimulation points: A—anode, K—cathode; G—ground electrode, LAT—latency; CV—segmental conduction velocity; AMP—amplitude; AREA—area of motor potential; DUR—duration of motor potential).

**Figure 4 sensors-20-02162-f004:**
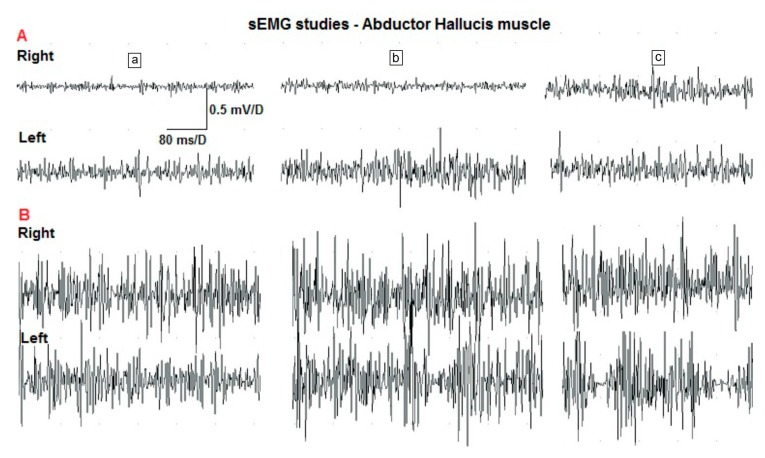
Sample sEMG results of a selected patient from study group A (a—the first phase of TSO exercises, b—the second phase of TSO exercise, c—the third phase of TSO exercise, right—recording from AbdH in the right foot, left—recording from AbdH in the left foot).

**Figure 5 sensors-20-02162-f005:**
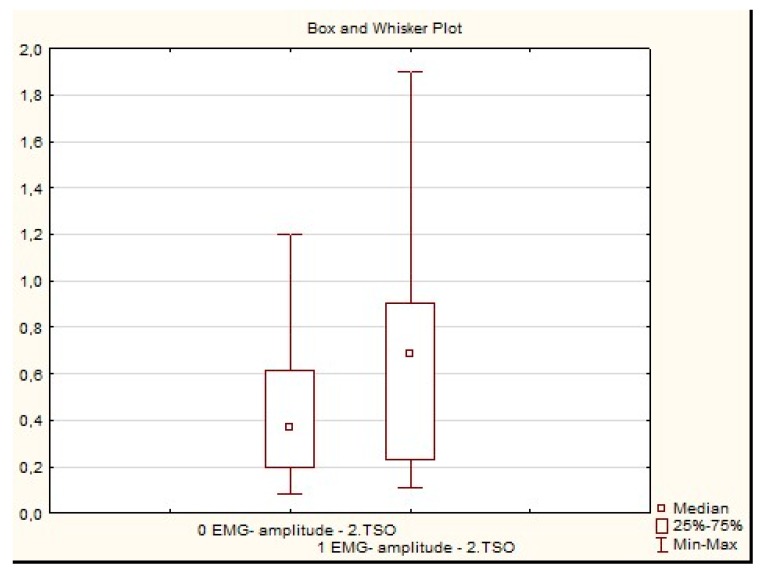
Comparison of the amplitudes in the second phase of toe-spread-out exercise in research group B (mV—millivolts, 2 TSO—the second phase of the toe-spread-out exercise, 0 EMG—examination before therapy, 1 EMG—examination after therapy).

**Table 1 sensors-20-02162-t001:** The characteristics of the valgus of the exercised foot according to the Manchester scale.

Manchester Scale	Number of patients	%
B	8	38.09524
C	11	52.38095
D	2	9.52381

**Table 2 sensors-20-02162-t002:** Results of comparison of the ENG parameters before and after therapy in research group A and B. A *p*-value below 0.05 was judged to be statistically significant.

	Compared Parameter	Statistical Test	*p*-Value
Research group A	Amp–MM ^1^	Student’s *t*-test	0.89
	Amp–PF ^2^	Student’s *t*-test	0.16
	Lat–MM ^3^	Student’s *t*-test	0.29
	Lat–PF ^4^	Wilcoxon test	0.05
	CV ^5^	Student’s *t*-test	0.06
Research group B	Amp–MM ^1^	Student’s *t*-test	0.85
	Amp–PF ^2^	Student’s *t*-test	0.25
	Lat–MM ^3^	Wilcoxon test	0.34
	Lat–PF ^4^	Student’s *t*-test	0.43
	CV ^5^	Wilcoxon test	0.42

^1^ amplitude measured below the medial malleolus, ^2^ amplitude measured in the popliteal fossa, ^3^ latency measured below the medial malleolus, ^4^ latency measured in the popliteal fossa and ^5^ conduction velocity.

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
