# Peer review of "Muscle Activity Detectors—Surface Electromyography in the Evaluation of Abductor Hallucis Muscle"

_sensors, 2020, doi:10.3390/s20082162_

Round 1

Reviewer 1 Report

Authors suggested an approach to assess the effect of the toe-spread-out exercise on the outcomes of sEMG recorded from the abductor hallucis muscle.  21 subjects were involved in this research. The paper addressed an interesting issue, however, there still exists some problems that need to be considered and solved by the authors.

  • Introduction section, authors should improve this section by adding more related work, showing the weakness in that research, why their method is important, etc.,
  • Authors should describe their methodology clearly and give justification for each step. Please, separate Experimental data, and Methodology are needed.
  • The description of the experimental setups is confusing, authors should clearly explain it.
  • The discussion section in this manuscript are not enough, please add more details and results.

Reviewer 2 Report

This paper demonstrate TSO exercises can result in changes of the frequency pattern of AbdH muscle in patients with hallux valgus and without deformity. Electromyography and electroneurography are good methods for assessing the toe spread out 300 exercise but a comprehensive assessment should include other tests as well. 

Problem

  1. It is like a clinical study to prove the TSO exercise is useful and you use the EMG and ENG to do the assessment. The innovation will be to create a assessment method and understand what is the effect of TSO. I am not sure if it is suitable for this journal (this journal focus on the sensors and certain new sensor algorithm and technology)
  2. Do you have any innovation in those EMG and ENG sensor, what will it be?
  3. Except the exercise, are there other treatment? Maybe in your experiment design, you can also test what are the other treatments' effect, and compare those to conclude which treatment is better?
  4. Line 121 5mm2 should be 5mm2

Round 2

Reviewer 1 Report

Authors have addressed all my comments.  

Reviewer 2 Report

The article is already modified.